# Biomechanical Consequences of Using Passive and Active Back-Support Exoskeletons during Different Manual Handling Tasks

**DOI:** 10.3390/ijerph20156468

**Published:** 2023-07-28

**Authors:** Mathilde Schwartz, Kévin Desbrosses, Jean Theurel, Guillaume Mornieux

**Affiliations:** 1Working Life Department, French National Research and Safety Institute for the Prevention of Occupational Accidents and Diseases (INRS), 54500 Vandœuvre-les-Nancy, France; 2Développement Adaptation et Handicap (DevAH), Université de Lorraine, 54000 Nancy, France; 3Faculty of Sport Sciences, Université de Lorraine, 54000 Nancy, France

**Keywords:** EMG, kinematics, handling tasks, wearable assistive devices, musculoskeletal disorders, low back pain

## Abstract

The aim of this study was to assess, for both men and women, the consequences of using different back-support exoskeletons during various manual material tasks (MMH) on the activity of back muscles and trunk kinematics. Fifteen men and fourteen women performed MMH involving a 15 kg load (a static task, a symmetric lifting task, and an asymmetric lifting task). Four exoskeleton conditions were tested: without equipment (CON) and with three exoskeletons passive (P-EXO), and active (A-EXO1 and A-EXO2)). The electromyographic activity of the lower trapezius (TZ), latissimus dorsi (LD), erector spinae (ES), gluteus maximus (GM), and biceps femoris (BF) muscles was recorded. Trunk kinematics were evaluated to provide average thoracic, lumbar, and hip angles. The use of the P-EXO decreased the activity of LD, GM, and BF from −12 to −27% (*p* < 0.01) compared to CON, mostly during the static task. The A-EXO1 and A-EXO2 reduced the muscle activity of all studied muscles from −7 to −62% (*p* < 0.01) compared to CON and from −10 to −52% (*p* < 0.005) compared to the P-EXO, independently of the modalities of the experimental tasks. A statistical interaction between the sex and exoskeleton was only observed in a few rare conditions. Occupational back-support exoskeletons can reduce trunk extensor muscle activity compared to no equipment being used. However, these reductions were modulated by the exoskeleton technology (passive vs. active), design (weight and anthropomorphism), and the modalities of the task performed (static vs. dynamic). Our results also showed that the active exoskeletons could modify the trunk kinematics.

## 1. Introduction

Musculoskeletal disorders, such as low back pain (LBP), have negative effects on workers by decreasing their quality of life, but also on employers by disorganizing the work activity, eventually leading to a decrease in performance (productivity and quality) involving increased absenteeism and turnover [1,2]. In this context, and to guarantee the health and safety of workers, companies are turning toward physical assistance devices, such as occupational exoskeletons, when collective preventive measures cannot be applied. While passive systems (assistance provided by deformable materials storing energy) were first deployed in companies, technological development has led to active systems (using robotic motors) now being available on the market.

Previous studies observed that the use of occupational back-support exoskeletons can decrease the activity of back muscles during handling tasks, but also led to kinematic alterations [3,4,5,6]. However, the magnitude of these effects seems to depend on the characteristics of the exoskeleton, such as its technology (passive versus active) [4,7] or even its design (textile versus rigid, for example) [8], as well as the characteristics of the tasks performed (e.g., static tasks involving the maintenance of trunk flexion versus dynamic tasks that put the trunk in motion) [4,9,10,11] and the population studied [9,11,12].

Thus, a recent review reported that passive exoskeletons decreased muscle activity of the erector spinae (ES) and biceps femoris (BF) on average by −18 and −19%, respectively, while the reduction reached on average −25% for the ES, −41% for the gluteus maximus (GM), and −5% for the BF when using active exoskeletons [4]. Moreover, Poliero et al. (2022) [7], who compared a passive and an active exoskeleton, observed that the reductions on the trunk muscle extensors obtained with the active system were more than double those induced by the passive system in both the dynamic and static tasks (41 vs. 16% and 22 vs. 8%, respectively). The authors explained these results via a higher level of assistance for the active system than the passive one. Although a higher level of assistance is probably a determining factor in the increased benefits provided by active exoskeletons, the design of the system could also play a role. Schwartz et al. (2021) [8] highlighted that the differences observed between two passive exoskeletons could be related to their textile or rigid structure. When used in a stoop technique, the textile system allowed for greater assistance due to a better anthropomorphic feature. In the case of active exoskeletons, the design could also have an influence on their consequences. An active back-support exoskeleton can be single-joint (Robomate, Germany), for example [13]) or multi-joint (Exoback, for example) [14]. This design characteristic could thus contribute to making the exoskeleton more or less anthropomorphic. However, to our knowledge, no study has evaluated this factor for active systems. It was also observed that the use of back-support exoskeletons could induce some kinematic alterations, which can contribute in part to the previously observed muscle activity modifications. For example, during handling tasks requiring trunk flexion in the sagittal plane, it had been reported that the decrease in spinal erector muscle activity was mainly due to a decrease in flexion/extension (−19%), rotation (−24%), and lateral flexion of the trunk (−30%) associated with the use of a passive back-support exoskeleton (PLAD, Canada)) [15]. However, kinematic alterations were not always observed for all systems. Some authors reported a decrease in knee flexion [6,16] with the use of two passive back-support exoskeletons, while others measured no difference with another exoskeleton [10,17,18]. Therefore, these differences in the kinematic alterations could partly be linked to different designs. However, in the current state, it is difficult to know which characteristics of the system (technology, design, etc.) influence the biomechanical consequences, since the differences observed between exoskeletons come mainly from different studies that often use various protocols. Therefore, it is necessary to compare several technologies within the same measurement protocol to better understand the influence of these wearable assistive devices.

In addition, previous studies have shown that the characteristics of the work task could also influence the biomechanical consequences of using exoskeletons [4,9,10,11]. For example, Luger et al. (2021) [10] reported that a passive exoskeleton can induce reductions in ES muscle activity during a symmetric dynamic task (−12%) compared to a static one (−19%). These authors also reported an increase in ES muscle activity (+1%) when using the exoskeleton during an asymmetric dynamic task. These findings support those by Alemi et al. (2020) [9], who reported a larger reduction in ES muscle activity during a symmetric task (29%) than an asymmetric task (9%) with the use of another passive exoskeleton (BackX, Emeryville, CA, USA)). These results underline that the biomechanical consequences of using a passive exoskeleton are probably task dependent. However, little is known about this task-dependency when using an active exoskeleton. Indeed, although different authors have reported decreased muscle activity when using an active exoskeleton, these measurements have mostly been carried out during dynamic tasks in the sagittal plane [4]. Therefore, the possible wider benefits of using active exoskeletons compared to passive ones cannot be fully understood until a comparison has been made during various handling tasks.

Finally, exoskeletons (active and passive) should help all workers, e.g., regardless of their sex. However, some previous studies highlighted that sex (men versus women) may influence the muscular activity modifications associated with the use of these systems [9,11,12]. For example, Alemi et al. (2020) [9] reported an interaction effect between passive exoskeletons and sex on ES muscle activity: BackX resulted in a significant reduction for both men and women, while Laevo V2.5, Delft, The Netherlands) yielded a reduction only for women. Furthermore, the relative reductions were higher for women (−21% for BackX and −23% for Laevo) than for men (−15% for BackX). The authors hypothesized that these results could be related to anthropomorphic differences, in particular the weight of the torso, which may be lighter in women. The latter would obtain additional benefits compared to men, explained by a similar level of assistance between the two sexes (defined by the exoskeleton) but an overall lighter weight to be assisted in women. However, these differences in terms of sex were not corroborated by others [8,19]. Schwartz et al. (2021) [8] observed similar changes in ES muscle activity for both men and women regardless of the passive exoskeleton used (Laevo and Corfor, Villemus, France)). Unfortunately, to our knowledge, no studies have considered the potential effect of sex on biomechanical parameters with active systems. Most studies have only evaluated exoskeletons in men, but there are biomechanical differences between the two sexes during MMH [20,21,22].

Considering these elements, the objective of this study was to evaluate the neuromuscular and kinematic consequences of the use of different back-support exoskeletons, distinguished by the technologies used (passive or active) and their design, in different standardized work tasks and with two different populations, to better understand the impact of these variables as well as their interactions. Our main hypothesis was that active exoskeletons would be more effective than passive exoskeletons in reducing back muscle activity and that the benefits would be dependent on the task performed.

## 2. Materials and Methods

### 2.1. Participants

Twenty-nine volunteers (15 men: 25 ± 5 years, 180 ± 4 cm, 74.9 ± 7.7 kg; 14 women: 24 ± 6 years, 166 ± 6 cm, 63.6 ± 13.3 kg), without any musculoskeletal pathologies, participated in this study. All participants completed a training protocol before the experiments began. This training consisted of five sessions of 2 hours, allowing the participants to familiarize themselves with the different exoskeletons used, as well as the experimental tasks performed in the study. Participants gave written consent after receiving detailed information about the objectives, protocol, and risks of the study. A national ethics committee approved the study (No. IDRCB 2019-A02901-56).

### 2.2. Experimental Design

Participants performed three standardized MMH: a static load holding task, a symmetric lifting task, and an asymmetric lifting task (described below). During the tasks, the position of the feet was fixed, parallel, and hip-width apart. This position was marked at the beginning of the experiment and kept identical for all conditions during the experiment. All participants wore safety shoes during the study. The order of the experimental conditions (task × exoskeleton) was randomized. Between each condition, a minimal rest period of 5 min was observed and also allowed participants to equip themselves with the exoskeletons.

#### 2.2.1. Simulated Work Tasks

The static task (Static_task) consisted of holding, for a duration of 10 s, a 15 kg load at knee height (Figure 1A). The participants maintained the load with the trunk flexed, the arms extended, and without locking the knee joint in full extension. The position was visually checked by the experimenters and adjusted if necessary. This task was repeated twice with a recovery period of at least 30 s.The symmetric lifting task (Sym_task) consisted of lifting a 15 kg load from a platform positioned at ankle level to a platform located at elbow level, and vice versa, at an imposed rate of 15 cycles/min while using a rhythmic beep (Figure 1B). Both platforms were located in front of the participants to impose a trunk flexion and extension in the sagittal plane. The higher platform was set back from the lower platform to require a slight elbow extension in the sagittal plane after the trunk extension. The height of the two platforms was adjusted to the anthropometric characteristics of the participants. Participants performed the task with a stoop technique. The working posture was visually checked by the experimenters and adjusted if necessary. During the task, the subjects did not deposit the load on the platforms but simply had to touch it. The participants completed two trials of five successive cycles (a cycle corresponds to the lifting and lowering of the load). A minimum 1-min rest period was observed between each trial.The asymmetric lifting task (Asym_task) was similar to the previous one, but with a high platform placed on the right-hand side of the participant (Figure 1C). Therefore, a 90° trunk rotation around the longitudinal axis to the right was imposed during lifting and lowering movements.

#### 2.2.2. Exoskeletons

One passive exoskeleton (P-EXO) and two active exoskeletons (A-EXO1 and A-EXO2) were studied (Figure 2) and compared to a control condition without any equipment (CON). These exoskeletons have already been studied by Schwartz et al. (2022) [14], and additional details about these assistive devices were provided in their study.

P-EXO: the BackX model (S model, 2019) was a passive exoskeleton from SuitX (Emeryville, CA, USA) (Figure 2A). The exoskeleton was adjusted to each participant’s morphology and was set to the “instant” mode with the “high” assistance level, corresponding approximately to a maximum torque of 41 Nm at a 90° trunk flexion. This assistance level was chosen according to the weight to be handled (15 kg).A-EXO1: the Exoback model (V1, 2020) is an active exoskeleton from RB3D (France) (Figure 2B). During the Static_task, the exoskeleton assistance of the trunk flexion (called “support level” in the settings) was individually adjusted so that the exoskeleton fully supported the weight of the trunk, and the load was handled in the desired static posture (this adjustment was made according to the participant’s feeling of being fully supported). Most of the participants could remain, without any effort, fully supported by the exoskeleton. For some participants (seven men and three women), the maximum assistance level (72 Nm) was not sufficient to feel fully supported. In this case, maximal assistance was selected in the settings. During dynamic tasks (Sym_task and Asym_task), the assistance level of the trunk extension was set to 100%, corresponding to 72 Nm. We chose this level according to the manufacturer’s recommendation with respect to the weight to be handled (15 kg). The reactivity as well as the softness parameters were set at 50% (default settings). The support level, level of assistance at the end of the movement, and hold threshold parameters were set to 20% (default settings). The exoskeleton was adjusted to each participant’s morphology.A-EXO2: the Cray X (2019) is an active exoskeleton from German Bionic (Augsburg, Germany). During the Static_task, the static mode was selected, and the assistance was individually set so that the weight of the participant’s trunk and the load handled were fully supported (based on the same criteria as above). For some participants (11 men and 4 women), the maximum level of assistance (120 Nm) was not sufficient to feel fully supported. In this case, maximal assistance was selected in the settings. During the dynamic tasks (Sym_task and Asym_task), the dynamic mode was selected with the assistance level set at 60%, corresponding to 72 Nm and comparable to A-EXO1. The sensitivity, reaction, and counter force parameters were set to 5 (default settings on a scale of 0–10). The exoskeleton was adjusted to each participant’s morphology.

### 2.3. Data Acquisition and Analyses

#### 2.3.1. Electromyography

A surface electromyography (EMG) system (Wave Plus™, Cometa, Bareggio, Italy) recorded the activity of 10 muscles involved in the trunk extension (considered as hip and spine extension): bilaterally, the biceps femoris (BF), gluteus maximus (GM), erector spinae (ES), latissimus dorsi (LD), and the lower trapezius (TZ) muscles. Bipolar electrodes (BlueSensor N-00-S, Ambu, Columbia, MD, USA) were placed on the skin according to the SENIAM recommendations [23]. The inter-electrode distance was 2 cm, and the impedance was controlled at less than 5 kΩ. The signal was recorded at 2000 Hz, amplified (×1000), and bandpass filtered from 10 to 500 Hz. A 4th-order highpass Butterworth filter at 30 Hz was also applied to remove the heart rate signal present on some EMGs [24].

Prior to performing the work tasks, two 5 s isometric maximal voluntary contractions (MVC) were performed successively for the BF, GM, LD and TZ muscles. For the ES muscles, the choice was made to perform submaximal isometric contractions, on the one hand to preserve the ES region from maximal efforts, and on the other hand, to have elements of comparison with a previous study that evaluated various passive exoskeletons in similar work tasks [8]. A rest period of 1 min was observed between the two MVCs. During the MVC, the RMS (Root Mean Square) was calculated over successive periods of 500 ms sliding windows in 0.5 ms steps, and the highest value was used as the reference value (RMS_REF_).

Next, during the dynamic work tasks (Sym_task and Asym_task), an RMS value was calculated for each muscle during the total time of each trunk extension (load lifting). The timing of the extension was defined using contact sensors placed on the platforms, to signal the beginning and the end of each movement. For the Static_task, an RMS value was calculated over 8 seconds during each trial. To avoid any disturbance of the movement linked to the start or end of the task, the first and last trunk extensions during the dynamic tasks, as well as the first and last seconds during the static task, were not selected for data analysis. Then, RMS values were expressed via percentages of RMS_REF_. Moreover, for the sagittal plane tasks (Static_task and Sym_task), the EMG activity of the right and left muscles was averaged.

#### 2.3.2. Kinematics

Three inertial measurement units (IMUs) (SEN-09268), each composed of a 2-axis gyroscope (IDG500) and a 3-axis accelerometer (ADXL335) were placed on the C7 cervical, T8 thoracic and L5 lumbar vertebrae. A triaxial accelerometer (ADXL335) was positioned laterally on the thigh, on the axis passing through the greater trochanter and the lateral femoral epicondyle. The signals of these sensors were sampled at 250 Hz and synchronized with the EMG data. An offset correction and a 2nd-order lowpass Butterworth filter at 3 Hz were applied to the kinematic data [25].

The inclination of each sensor (C7, T8, L5, and thigh) relative to the axis of gravity was calculated. For the thigh (almost static body segment during the three tasks), the inclination was calculated directly from the accelerometric values. For C7, T8, and L5 (more dynamic body segments), the inclination was calculated using a fusion algorithm that considered coefficients of 0.99 for the gyroscope data and 0.01 for the accelerometric data [25]. Then, the difference in inclination between C7-T8, T8-L5, and L5-thigh was calculated to obtain the thoracic, lumbar, and hip angles, respectively. Data were analyzed and averaged during each trunk extension for the Sym_task and Asym_task and over 8 s for the Static_task. To avoid any disturbance of the movement linked to the start or end of the task, the first and last trunk extensions during the dynamic tasks, as well as the first and last seconds during the static task, were removed from the analysis.

### 2.4. Statistical Analysis

The EMG (TZ, LD, ES, GM, and BF) and kinematic values (thoracic, lumbar, and hip angles) are presented as means ± standard deviations. A generalized linear mixed model was used to analyze the main effects of the exoskeleton (CON, P-EXO, A-EXO1, and A-EXO2) and sex (men and women), as well as their interaction effects, on the EMG and kinematic values during each task (Static_task, Sym_task, and Asym_task). The exoskeleton and sex independent variables were set as fixed effects and participants as the random effect. In case of a non-significant interaction, this was removed from the model. Significant effects were then analyzed using Bonferroni post hoc comparison tests. The significance level was set at 5% (*p* < 0.05). The effect size was evaluated by means of Cohen’s d values for post hoc comparisons. The effect size was considered small (0.2 ≤ d < 0.5), medium (0.5 ≤ d < 0.8), or large (d ≥ 0.8) according to Cohen [26]. In case of statistical significance (*p* < 0.05), only results with a small, medium, or large effect size (d ≥ 0.20) were presented. All analyses were performed using STATA software (Stata 16, StataCorp, College Station, TX, USA).

## 3. Results

### 3.1. Electromyography

The EMG results are presented in two parts: one is specific to the ES muscles, mainly investigated in previous studies on back-support exoskeletons, and the other concerns all other muscles involved in the trunk extension (TZ, LD, GM, and BF).

#### 3.1.1. ES Muscles

##### Exoskeleton Effect

A significant main effect of the exoskeleton (*p* < 0.05) was observed in ES muscle activity during the three experimental tasks (Static_task, Sym_task, and Asym_task) (Figure 3). The active exoskeletons (A-EXO1 and A-EXO2) induced lower (*p* < 0.01) ES muscle activity compared to no equipment being used (CON) for all tasks. On average, for all these tasks, the active exoskeletons (A-EXO1 and A-EXO2) induced a relative decrease of 12–45% (*p* < 0.01) on ES muscle activity compared to the control situation without any equipment. No difference was observed between the passive exoskeleton (P-EXO) and CON.

Moreover, during the Static_task, ES muscle activity was lower for A-EXO2 compared to P-EXO (−35% in relative reduction, *p* = 0.005). During the Sym_task and Asym_task, ES muscle activity was lower for A-EXO1 and A-EXO2 compared to P-EXO (from −10 to −15%, *p* < 0.001).

##### The Effect of Sex and Interaction

A significant main effect of sex was only observed in the right ES muscle activity during the Asym_task, with higher values (*p* = 0.02) for women (84.9 ± 22.6%) than for men (67.8 ± 21.1%). Finally, there was no interaction effect between the exoskeleton and sex on ES muscle activity during the three experimental tasks.

#### 3.1.2. Other Muscles

##### Exoskeleton Effect

A significant main effect of the exoskeleton was observed for the other muscles studied (TZ, LD, GM, and BF) in the different tasks performed (*p* < 0.05, Table 1).

During the Static_task, the muscle activities were lower for the three exoskeletons compared to CON, except for TZ with P-EXO. The relative reduction in muscle activities ranged from −13 to −27% with the use of P-EXO (*p* < 0.01) and from −35 to −62% with A-EXO1 and A-EXO2 (*p* < 0.001). The muscle activity of all muscles was also lower with A-EXO1 and A-EXO2 compared to P-EXO (from −24 to −52%, *p* < 0.001).

During the Sym_task, there was a reduction in LD muscle activity for P-EXO compared to CON (−20%, *p* < 0.001). Both active exoskeletons also induced a lower activity on all muscles compared to CON (from −19 to −40%, *p* < 0.001) and to P-EXO (from −13 to −30%, *p* < 0.001).

Regarding the Asym_task, the left GM muscle activity was reduced with P-EXO1 compared to CON (−12%, *p* < 0.001). The two active systems produced lower muscle activity for all muscles compared to CON (from −7 to −35%, *p* < 0.001) and P-EXO (from −12 to −30%, *p* < 0.001), except for the left BF muscle. Finally, a reduced left GM muscle activity was observed for A-EXO1 compared to A-EXO2 (−16%, *p* < 0.001).

##### The Effect of Sex and Interaction

A significant main effect of sex was observed for TZ, LD, GM, and BF muscle activity. Values were higher for women than for men in the three tasks (from +32 to +164%, *p* < 0.05), except for the TZ muscle during the Static_task and Asym_task.

The interaction effect between exoskeletons and sex was found to be significant for some muscles only and never consistent across all the tasks performed. In the Static_task, LD muscle activity was lower for P-EXO compared to CON for women only (−39%, *p* < 0.001), and the GM muscle activity was lower for A-EXO1 compared to A-EXO2 for men only (−16%, *p* < 0.05).

For the Sym_task, TZ muscle activity was lower for A-EXO1 compared to A-EXO2 for men only (−10%, *p* < 0.05), and the BF muscle activity was lower for P-EXO compared to CON for women only (−8%, *p* < 0.001).

For the Asym_task, there were slightly more differences. The right TZ muscle activity was lower for A-EXO1 compared to A-EXO-2 for men (−18%, *p* < 0.001), and, conversely, for women (+14%, *p* < 0.001). The left LD muscle activity was reduced for P-EXO compared to CON for women only (−10%, *p* < 0.001). The left GM muscle activity was lower for P-EXO compared to CON for men (−17%, *p* < 0.001) and between A-EXO2 and P-EXO for women (−18%, *p* < 0.001). Finally, the left BF muscle activity was lower for A-EXO1 and A-EXO2 compared to CON for women (−11%, *p* < 0.001), but higher for A-EXO2 compared to CON, P-EXO, and A-EXO1 for men (+19, +15%, and +17%, respectively, *p* < 0.001).

### 3.2. Kinematics

#### 3.2.1. Exoskeleton Effect

The statistical model revealed a significant main effect of the exoskeleton on all angles in each task, except for the thoracic angle in both dynamic tasks (Sym_task and Asym_task) (Table 2). In the Static_task, the mean thoracic flexion angle was higher for A-EXO1 compared to CON (+6°, *p* < 0.001) and P-EXO (+5°, *p* < 0.001). Furthermore, A-EXO2 induced a lower lumbar flexion angle compared to CON (−4°, *p* = 0.02) and P-EXO (−5° *p* = 0.002), as well as a higher hip flexion angle compared to CON (+9°, *p* < 0.001) and A-EXO1 (+9°, *p* < 0.001).

During the Sym_task, a greater lumbar extension angle was found with A-EXO2 compared to the other conditions (CON: +8°, P-EXO +10°, and A-EXO1: +7°, *p* < 0.001). The hip flexion angle was also greater for A-EXO2 (*p* < 0.001) compared to CON (+11°), P-EXO (+7°), and A-EXO1 (+9°).

During the Asym_task, a higher lumbar extension angle was observed with A-EXO1 (+5°, *p* = 0.004) and A-EXO2 (+6°, *p* < 0.001) compared to P-EXO. A-EXO2 also induced a greater hip flexion angle compared to the other three conditions (CON: +10°, P-EXO: +8°, and A-EXO1: +7°, *p* < 0.001).

#### 3.2.2. Sex Effect and Interaction

A significant main effect of sex was only found for the two dynamic tasks (Sym_task and Asym_task). In the Sym_task, women had a greater thoracic flexion angle (+4°, *p* = 0.049) and a greater lumbar extension angle (+16°, *p* = 0.015) than men. For the Asym_task, the lumbar extension and hip flexion angles were greater for women than for men (+13°, *p* < 0.05).

The interaction effect between the exoskeleton and sex was found to be significant for the two dynamic tasks only. In the Sym_task, P-EXO resulted in a lower thoracic flexion angle compared to CON (−12°) and A-EXO2 (−10°) for women only (*p* < 0.001). In the Asym_task, the hip flexion angle was higher for P-EXO (+6°, *p* = 0.009) and A-EXO1 (+6°, *p* = 0.005) compared to CON for men only. For women only, it was observed that A-EXO2 produced a greater hip flexion angle than P-EXO (+15°) and A-EXO1 (+12°) (*p* < 0.001).

## 4. Discussion

This study investigated the influence of different back-support exoskeletons (one passive and two active) during various MMH (static, symmetric lifting, and asymmetric lifting) on trunk extensor muscle activity and trunk kinematics in both men and women. Our results highlighted that the use of the exoskeletons, compared to using no equipment, could reduce the erector spinae muscle activity but also that of other trunk extensor muscles. However, the magnitude of these reductions was dependent on the exoskeleton used and the task performed, with greater benefits for the two active systems. Our results also showed that only the active exoskeletons modified trunk kinematics. Finally, there was no consistent difference in the biomechanical effects induced by the use of the exoskeletons between men and women.

### 4.1. Consequences of Using a Passive Exoskeleton

Compared to using no equipment, the use of the passive exoskeleton (P-EXO) did not decrease the erector spinae muscle activity during the three tasks, but some reductions for other trunk extensor muscles, mainly during the static task, were observed. The lack of reduction in erector spinae muscle activity had previously been observed by Schwartz et al. (2021) [8], during a dynamic lifting task in the sagittal plane with another passive exoskeleton (Laevo V1). However, this result is not supported by others [8,10,11,17]. For example, Madinei et al. (2020) [11] observed a lower EMG activity during the flexion (−17%) and extension (−10%) of the trunk with the use of the same BackX exoskeleton. However, the mass of the load handled was much lower in their study, i.e., about 7 kg (corresponding to 10% of the participant’s body weight), compared to 15 kg in our protocol. This supports Abdoli et al. (2006) [27], who reported that increasing the load handled could reduce the benefits provided by passive systems. Thus, the conflicting results with Madinei et al. (2020) [11] could be due to the mass of the load being handled. Differences with other studies [8,10,17] could be attributed to the experimental tasks but also to the exoskeletons used. Indeed, it had previously been shown that for the same lifting task, conflicting results existed between two passive exoskeletons, with a reduction in erector spinae muscle activity for one system, but a lack of effect for the other [8].

But beyond the erector spinae muscles, our study reported reductions in the activity of other trunk extensor muscles. While the use of P-EXO did not induce significant decreases during the dynamic tasks, reductions in the muscle activity of the latissimus dorsi (−27%), gluteus maximus (−13%), and biceps femoris (−14%) were reported during the static one. This result confirms the fact that it is necessary to consider all trunk extensor muscles and not only the erector spinae muscles, as proposed in the previous study by Schwartz et al. (2021) [8]. These muscle activity reductions support the literature [8,9,10,17]. For example, Luger et al. (2021) [10] observed decreases of −8% and −36% in the biceps femoris muscle activity when using a passive system (Laevo) during a static and dynamic task, respectively. Bosch et al. (2016) [17] also reported, during a static task performed with the previous exoskeleton, a −24% reduction in the biceps femoris muscle activity.

Overall, these neuromuscular changes are probably related to both the characteristics of the exoskeletons used and the tasks performed. Indeed, it has been shown that the level of assistance of a passive exoskeleton could be variable during a lifting movement, depending on the direction of the movement (flexion versus extension of the trunk) and the trunk inclination angle [28]. So, the passive systems could provide optimal assistance during specific phases of the movement only. As the trunk extensor muscles also have an activation pattern dependent on the angle of trunk inclination [29], it seems that the assistance profile of a passive exoskeleton may match the activation phase of one or more muscles. It therefore seems important to investigate different muscles to ensure the global neuromuscular benefits of the exoskeletons. Regarding the kinematic data, our results highlighted similar trunk angles (thoracic, lumbar, and hip) between the use of a passive exoskeleton and without equipment during the three tasks. Madinei et al. (2020) [11] also observed that the range of motion during a dynamic MMH was not modified using the same passive exoskeleton. However, another study pointed out that the use of this passive exoskeleton could induce some substantial modifications, such as a decrease in the lumbar flexion of about 5° during a static task [30]. Thus, based on previous studies and our metrics used (average angle), the use of BackX does not seem to induce major changes in overall trunk kinematics. The muscular benefits observed during the use of this system is probably related to its assistance and not to any modification of the kinematics as suggested in the literature [15]. More in-depth studies of whole-body kinematics would be interesting to confirm these results.

### 4.2. Consequences of the Use of Active Exoskeletons

Compared to using no equipment, the use of the two active exoskeletons (A-EXO1 and A-EXO2) induced a reduction in EMG activity from −12 to −45% for the erector spinae muscle and from −7 to −62% for the other trunk extensor muscles (biceps femoris, gluteus maximus, latissimus dorsi, and lower part of the trapezius) during all tasks performed. These results corroborate previous studies conducted with other active exoskeletons and during different work tasks [7,13,31,32,33], even if these studies mainly focused on the erector spinae muscle. Moreover, our results yielded a comparison between different exoskeletons and highlighted, for the first time, greater reductions in EMG activity for all trunk extensor muscles with both active exoskeletons compared to the passive one, throughout the different tasks. This confirms the literature review by Kermavnar et al. (2021) [4], which indicated an average decrease of −25% in erector spinae activity with active systems and about −18% with passive systems compared to no equipment. These differences between the two technologies (passive vs. active) could be explained via a higher level of assistance provided by active exoskeletons [7]. In our study, during the two dynamic tasks, the level of assistance of the active systems was set according to the manufacturers’ recommendations for handling a 15 kg load: the active exoskeletons provided a maximal torque of 72 Nm while the passive one only achieved a maximal torque of ≈41 Nm (highest setting with this device). For the static task, the torque setting was also always higher with the active exoskeleton compared to the passive one. Thus, in our study, the active systems induced additional benefits for the trunk extensor muscles, likely due to a higher level of assistance.

Regarding the comparison of the two active exoskeletons (A-EXO1 vs. A-EXO2), it was observed that the muscle activity was similar, except during the asymmetric lifting task where the left gluteus maximus muscle activity was lower (−16%) for A-EXO1 compared to A-EXO2. This difference alone does not allow us to distinguish a real difference between these two systems in terms of physical workload and prevention of lower back pain. Furthermore, it was observed that the effects were dependent on the task performed. Indeed, during the static task, the use of active exoskeletons induced an average −46% reduction in trunk extensor muscle activity whereas, during the dynamic tasks, decreases were on average −23%. This finding is in agreement with what has been observed in the literature: a −39% decrease in erector spinae muscle activity during a static task [33] and only −13% during dynamic tasks [13] even though different exoskeletons were used in these studies. The increased efficiency of the active exoskeleton used in this study could be related to the settings of the assistance and the production thereof. Indeed, the active systems enabled different settings according to the task performed: a “static mode” allowed support for a static trunk flexion position and a “dynamic mode” generated torque upon detection of a trunk extension movement. Thus, during the static task, users were continuously assisted by the exoskeletons, which fully supported the overall mass (trunk + load) in order to maintain the position (by using maximal torque). During the dynamic tasks, the assistance may be less efficient as the systems had to recognize the movement first (which can take some time according to the control laws) and “follow” it to try to aid the users throughout the movement. In this “dynamic” setting, the torque generated by the exoskeleton might not be the maximum over the whole range of motion, as already observed for passive systems [28], even if no information is available about the torque/angle relationship of these two active exoskeletons. It is also possible that there could be a loss of contact and/or less contact between the assistance transfer points of the exoskeletons and the user, therefore limiting the efficiency of the exoskeleton, especially if the user was moving faster than the system. In particular, this phenomenon could occur during the initiation of the trunk extension movement, such as the hysteresis effect already observed for passive exoskeletons [28]. Overall, this might also explain the different benefits provided to users during movements compared to maintained postures. However, further studies are necessary to validate these hypotheses, especially during the initiation of the movement, for which settings are available in terms of motion detection sensitivity and/or system reactivity.

It was also observed that the use of these two active systems, contrary to the passive one, could modify trunk kinematics. The different designs of the exoskeletons could explain these alterations. For example, the use of A-EXO1 (Exoback) induced a larger thoracic flexion compared to P-EXO (+6°) and CON (+5°) during the static task. This system had a different mechanical conception, especially in terms of the assistance transfer points at shoulder level and on the buttocks and thighs, with two joints (one on the hips and the other on the lower back), and no contact with the spine (the box being offset from the back). During the static task, this design would have allowed the user to find support by rolling up the spine, inducing a more significant thoracic flexion angle. The other active system (A-EXO2, CrayX) induced a reduction in the lumbar flexion from −4 to −10° and an increase in hip flexion of about 10° compared to the other assistive conditions. The rigid structure, in contact with the back, as well as the assistance transfer point at thorax level and the single joint on the hips, may have limited lumbar flexion and promoted hip flexion during the trunk inclination. The differences in kinematics observed between the two active back-support exoskeletons may then have generated modifications in the trunk extensor muscle activity. However, the benefits provided to the muscles were similar.

### 4.3. Influence of the User’s Sex

Overall, women had higher levels of muscle activity than men. This result may be explained by the fact that all participants carried the same load (15 kg) in this study, whereas women generally have a lower strength than men, on average [20,22]. The results might have been different if the load handled had been adapted to each participant’s maximal force. However, with the same load for all, we are more representative of what can happen in real-work situations. These differences in muscle activity could also be explained, in part, by the fact that men and women do not have the same trunk kinematics during MMH [21]. Indeed, in this study, greater lumbar extension and/or hip flexion angles were observed in women compared to men during dynamic tasks, confirming that women lean forward by mobilizing more hip flexion, while men mobilize more back curvature during a lifting task [21].

Moreover, the exoskeleton model induced some significant differences between men and women. However, these differences were not consistent across the different muscles and tasks and never influenced the erector spinae muscle activation. Thus, our results do not identify an exoskeleton technology or design that is more appropriate, regarding female or male workers. Nevertheless, it would seem that P-EXO (BackX) and A-EXO2 (CrayX) allowed women to stand out from men in reducing the activity of certain muscles in some tasks. In parallel, for men, P-EXO (BackX) and A-EXO1 (ExoBack) tended to be different, by comparison with women, for certain (other) muscles and (other) tasks. A match between the kinematics of the exoskeleton and the trunk kinematics of the user could explain these distinctions. Indeed, A-EXO2 has a single joint (at hip level) and a box in contact with the back: it therefore mainly promotes hip flexion within trunk inclination, which women tend to do without an exoskeleton [21]. Conversely, A-EXO1 has two joints (on the hips and on the lower back) and a box offset from the back: it therefore allows for a larger curvature of the back, usually used by men without any equipment [21]. As for P-EXO, this is a compromise between the two other systems, with a single joint but no structure on the back. It therefore allows men and women to move closer to their natural kinematics. However, these hypotheses remain to be confirmed with other back-support exoskeletons.

### 4.4. Limitation

The tasks performed in this study were standardized and performed in a laboratory. The extrapolation of the results to more complex work tasks achieved in a real situation can become difficult because of many other factors to be considered (environmental, organizational, etc.). However, the control of the experimental conditions allowed us to limit or/and avoid the influence of these other factors as much as possible and focus on the consequences induced by the only factors we wanted to analyze (exoskeleton, task, and sex). The experiments were also conducted with a sample of young and healthy participants. Generalization to all workers seems therefore premature, even if the factor of sex had been included in the model. The question of the use of these exoskeletons for older workers or for people who have already been confronted with health problems, such as LBP, can be raised, for example. Muscular and kinematic adaptations may not be similar due to possible functional limitations [34]. Finally, this study focused on trunk extensor activation and trunk kinematics, while the use of active systems could yield consequences beyond the assisted muscles and/or joints, as demonstrated for passive systems [5,6]. This point is very important in the prevention of occupational accidents and diseases, where one of the fundamentals is to not displace the risk. Further studies are therefore needed to better understand all the consequences of using passive or active back-support exoskeletons.

## 5. Conclusions

In conclusion, this study first showed that it is necessary to investigate several trunk extensor muscles and not only the erector spinae muscle when evaluating a back-support exoskeleton. Indeed, depending on the assistance characteristics of the exoskeleton used, not all muscles could be assisted. It has also been observed that both passive and active back-support exoskeletons can provide a reduction in trunk extensor muscle activity during manual handling tasks (MMH). However, active exoskeletons, especially because of a higher level of assistance, provide greater benefits. Although the two models of active exoskeletons induced trunk kinematic differences, the benefits in terms of the reduction in muscle activity were similar. Finally, the fact that these active systems seemed to perform better in static rather than dynamic conditions suggests that the assistance settings, although beneficial, were not optimal in these last conditions. For prevention purposes, it is therefore necessary to clearly identify the work task requiring assistance (static/dynamic, load handled, etc.) and to combine it with an appropriate exoskeleton whose performance must be known.

## Figures and Tables

**Figure 1 ijerph-20-06468-f001:**
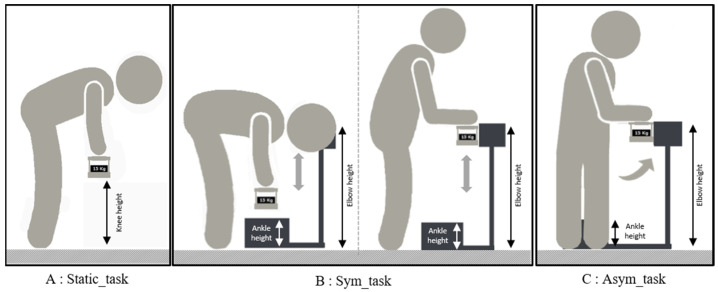
The three experimental tasks. (**A**) Static_task: the load was maintained at knee height; (**B**) Sym_task: from low position with load at ankle level to high position with load at elbow level; (**C**) Asym_task: the low position and load heights were identical to the Sym_task, but the high position involved a 90° trunk rotation on the longitudinal axis.

**Figure 2 ijerph-20-06468-f002:**
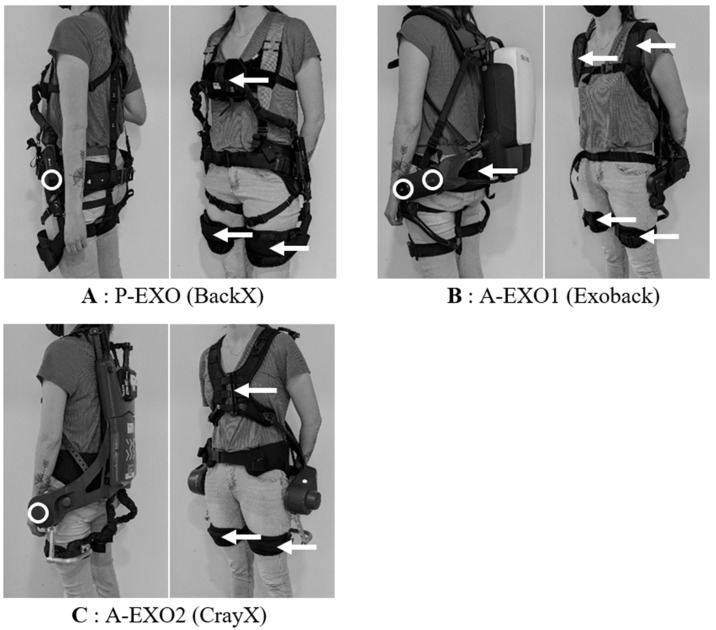
The three back-support exoskeletons. (**A**) P-EXO (BackX, Suit X), a passive exoskeleton. (**B**) A-EXO1 (Exoback, RB3D), an active exoskeleton. (**C**) A-EXO2 (Cray X, German Bionic), another active exoskeleton. Mechanical joints of the exoskeletons are represented by white circles, and points of assistance transfer are indicated by white arrows. Illustration reproduced and adapted from Schwartz et al. (2022) [14] with permission from Springer Nature.

**Figure 3 ijerph-20-06468-f003:**
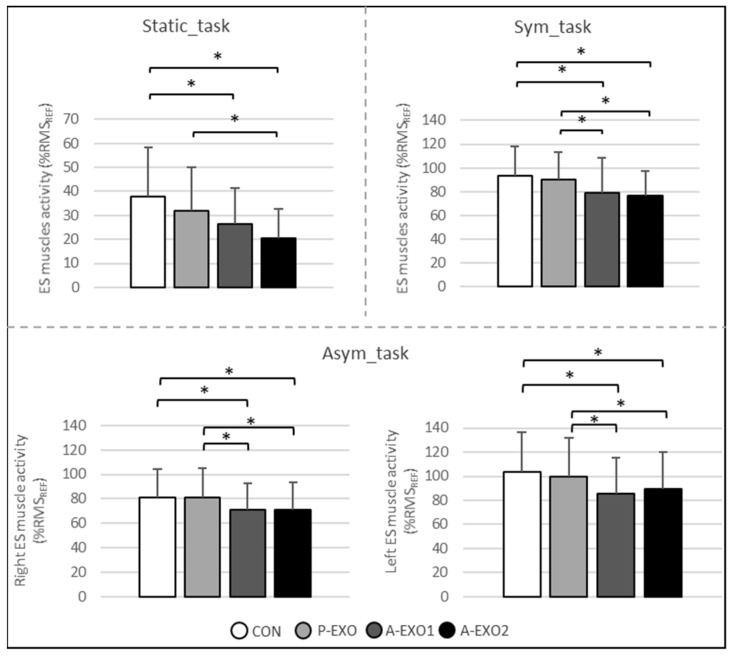
Mean EMG activities ± standard deviations of erector spinae (ES) muscles during the Static_task, Sym_task, and Asym_task. Values are expressed in %RMS_REF_. The control condition (CON, without equipment) is presented in white, the BackX passive exoskeleton (P-EXO) in light gray, the Exoback active exoskeleton (A-EXO1) in dark gray, and the CrayX active exoskeleton (A-EXO2) in black. A significant difference (*p* < 0.05 and d ≥ 0.2) is represented by a bracket with the symbol *.

**Table 1 ijerph-20-06468-t001:** Mean EMG activities ± standard deviations of trapezius (TZ), latissimus dorsi (LD), gluteus maximus (GM), and biceps femoris (BF) muscles during the Static_task, Sym_task, and Asym_task. Values are expressed in %RMS_REF_. CON: control condition (without equipment), P-EXO: BackX passive exoskeleton, A-EXO1: Exoback active exoskeleton, A-EXO2: CrayX active exoskeleton. * Significantly different to CON (*p* < 0.05 and d ≥ 0.2). # Significantly different to P-EXO. $ Significantly different to A-EXO1.

	**Static_task**						
	**CON**	**P-EXO**		**A-EXO1**		**A-EXO2**	
TZ	53.5 ± 39.9	49.0 ± 35.9		23.3 ± 30.4	* #	24.0 ± 22.0	* #
LD	13.0 ± 8.7	9.4 ± 7.8	*	4.9 ± 4.3	* #	5.0 ± 4.2	* #
GM	14.7 ± 8.0	12.7 ± 7.5	*	8.2 ± 5.7	* #	8.9 ± 5.4	* #
BF	26.4 ± 15.2	22.7 ± 13.1	*	16.4 ± 10.5	* #	17.0 ± 10.8	* #
	**Sym_task**						
	**CON**	**P-EXO**		**A-EXO1**		**A-EXO2**	
TZ	107.4 ± 70.5	114.5 ± 81.7		80.1 ± 57.9	* #	82.2 ± 58.5	* #
LD	19.4 ± 12.4	15.4 ± 10.0	*	12.3 ± 8.6	* #	11.5 ± 7.8	* #
GM	31.2 ± 14.1	29.7 ± 13.4		22.0 ± 9.6	* #	24.2 ± 11.7	* #
BF	37.7 ± 19.7	35.1 ± 18.9		30.0 ± 15.1	* #	30.3 ± 14.1	* #
	**Asym_task**						
	**CON**	**P-EXO**		**A-EXO1**		**A-EXO2**	
Right TZ	99.4 ± 37.1	99.9 ± 37.4		69.2 ± 29.4	* #	70.0 ± 25.4	* #
Right LD	24.0 ± 15.0	21.5 ± 14.2		15.5 ± 9.2	* #	16.7 ± 12.0	* #
Right GM	26.2 ± 15.2	24.7 ± 13.7		20.1 ± 12.4	* #	20.2 ± 11.0	* #
Right BF	35.3 ± 20.2	32.0 ± 18.7		26.0 ± 14.0	* #	25.9 ± 12.7	* #
Left TZ	97.4 ± 91.2	94.0 ± 87.7		68.6 ± 73.4	* #	70.4 ± 72.8	* #
Left LD	17.9 ± 15.4	15.9 ± 14.0		12.7 ± 10.3	* #	12.5 ± 10.1	* #
Left GM	36.2 ± 15.9	31.9 ± 15.4	*	23.9 ± 12.5	* #	27.8 ± 13.3	* # $
Left BF	37.1 ± 18.8	36.1 ± 17.6		34.4 ± 16.8		37.2 ± 18.8	

**Table 2 ijerph-20-06468-t002:** Mean ± standard deviations of the different joint angles (thoracic, lumbar, and hip angles in °) during the Static_task, Sym_task, and Asym_task. Positive values represent flexion angles. CON: control condition (without equipment), P-EXO: BackX passive exoskeleton, A-EXO1: Exoback active exoskeleton, A-EXO2: CrayX active exoskeleton. * Significantly different to CON (*p* < 0.05 and d ≥ 0.2). # Significantly different to P-EXO. $ Significantly different to A-EXO1.

	**Static_task**					
	**CON**	**P-EXO**	**A-EXO1**		**A-EXO2**	
Thoracic angle	43 ± 13	44 ± 14	49 ± 13	* #	47 ± 14	
Lumbar angle	18 ± 18	19 ± 14	18 ± 15		14 ± 15	* #
Hip angle	49 ± 21	53 ± 19	49 ± 19		58 ± 25	* $
	**Sym_task**					
	**CON**	**P-EXO**	**A-EXO1**		**A-EXO2**	
Thoracic angle	41 ± 19	35 ± 18	40 ± 18		41 ± 16	
Lumbar angle	−1 ± 22	1 ± 25	−2 ± 21		−9 ± 26	* # $
Hip angle	53 ± 20	57 ± 22	55 ± 21		64 ± 28	* # $
	**Asym_task**					
	**CON**	**P-EXO**	**A-EXO1**		**A-EXO2**	
Thoracic angle	37 ± 17	36 ± 15	37 ± 13		37 ± 14	
Lumbar angle	1 ± 20	3 ± 22	−2 ± 26	#	−3 ± 25	#
Hip angle	54 ± 18	56 ± 18	57 ± 25		64 ± 26	* # $

## Data Availability

The data presented in this study are available on request from the corresponding author. The data are not publicly available due to ethical regulations.

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
