# Peer review of "Biomechanical Consequences of Using Passive and Active Back-Support Exoskeletons during Different Manual Handling Tasks"

_ijerph, 2023, doi:10.3390/ijerph20156468_

Round 1
Reviewer 1 Report
General comments:
This manuscript reports a study examining differences in trunk muscle activities and trunk kinematics during lifting tasks performed by men and women with and without exoskeletons. The authors concluded that occupational back-support exoskeletons can lower trunk extensor muscle activities and the level of reductions in muscle activities depends on the modalities of the task and the design of the exoskeleton. Also authors indicated that using exoskeletons could result in changes in trunk kinematics.
This paper was well written. The reviewer has some suggestions that could further improve the quality and clarity of the paper. Firstly, readers have different backgrounds, not all of them are familiar with the terms of static and dynamic tasks. Please briefly explain the nature of the static and dynamic tasks in the introduction before using them later. Secondly, similar to the first suggestion, for readers with limited knowledge of the exoskeleton technology, the term passive and active devices could be defined in the introduction to bring them up to speed of understanding the paper. They don’t need to jump to the method section to know the general specifications of the devices.
Specific comments:
Method:
In 2.3.1. Electromyography section, please indicate the window size of the RMS procedure. Is it 0.1 second?
Results:
In 3.2.1. table 2, the numbers of the Sym_task and Asym_task are identical.
Author Response
The authors would first like to warmly thank the reviewers for the time spent in reading our manuscript and providing comments that improve the text. We hope that the answers provided below and the amendments made in the manuscript (in red) will answer your concerns.
Reviewer 1
Comments and Suggestions for Authors
General comments:
This manuscript reports a study examining differences in trunk muscle activities and trunk kinematics during lifting tasks performed by men and women with and without exoskeletons. The authors concluded that occupational back-support exoskeletons can lower trunk extensor muscle activities and the level of reductions in muscle activities depends on the modalities of the task and the design of the exoskeleton. Also authors indicated that using exoskeletons could result in changes in trunk kinematics.
This paper was well written. The reviewer has some suggestions that could further improve the quality and clarity of the paper. Firstly, readers have different backgrounds, not all of them are familiar with the terms of static and dynamic tasks. Please briefly explain the nature of the static and dynamic tasks in the introduction before using them later. Secondly, similar to the first suggestion, for readers with limited knowledge of the exoskeleton technology, the term passive and active devices could be defined in the introduction to bring them up to speed of understanding the paper. They don’t need to jump to the method section to know the general specifications of the devices.
Thank you. Concerning the explanation of the nature of the static and dynamic tasks and the term passive and active devices, we added additional information (in brackets) in the introduction. Line 39: “While passive systems (assistance provided by deformable materials storing energy) were first deployed in companies, technological development has led to active systems (using robotic motors), now available on the market”. Line 44: “However, the magnitude of these effects seems to depend on the characteristics of the exoskeleton, such as its technology (passive versus active) [2, 5] or even its design (textile versus rigid for example) [6], as well as the characteristics of the tasks performed (e.g. static tasks involving the maintenance of specific trunk flexion angle or dynamic tasks that put the trunk in motion) [2, 7-9] and the population studied [7, 9, 10]”.
Specific comments:
Method:
In 2.3.1. Electromyography section, please indicate the window size of the RMS procedure. Is it 0.1 second?
Thank you for this feedback. We modified line 233 in “During the MVC, the RMS (Root Mean Square) was calculated over successive periods of 500-ms sliding windows in 0.5-ms steps and the highest value was used as the reference value (RMSREF)”. Concerning the different movements studied, we calculated average RMS over the full time of the task. For the static task, the calculation of the RMS has been performed over the middle 8 seconds of the task, as specified in the text. For the dynamic tasks (Sym_task and Asym_task), the calculation of the RMS has been performed over the total time of the trunk extension. This time was determined by the contact sensors placed on the platforms. We have modified the sentence line 237: “Next, during the dynamic work tasks (Sym_task and Asym_task), an RMS value was calculated for each muscle during the total time of each trunk extension (load lifting).”
Results:
In 3.2.1. table 2, the numbers of the Sym_task and Asym_task are identical.
We apologize for this mistake. We made the correction in the table 2.
Reviewer 2 Report
Dear authors,
congratulations for the good work done.
I have just some minor concerns about the article.
INTRODUCTION:
- references to MSDs and LBP incidence and their consequences need to be included.
- when citing a study use LastNameFirstAuthor + et al., + (PubblicationYear).
MATERIALS AND METHODS
Exoskeletons:
- for the two active exoskeletons, it is not clear what really are all the different setting parameters and their defualt values (reactivity, softness, hold threshold parameters for the first device, sensitivity, reaction and counter force for the second).
"The reactivity as well as the softness parameters were set at 50% (default settings). The support level, level of assistance at the end of the movement, and hold threshold parameters were set to 20% (default settings)."
"The sensitivity, reaction and counter force parameters were set to 5 (default settings on a scale of 0–10)."
Both these sentences don't make sense for a reader who doesn't know these specific devices.
I suggest explaining in detail these settings only if you can provide a clear explanation (e.g., what sensitivity is, what the value 5 is).
So maybe it is better if you explain only the level of assistance.
- Figure 2: Make it bigger and colored to make the exoskeletons more visible.
RESULTS
- The EMG results are presented in two parts: one is specific to the ES muscles, mainly investigated by previous studies on back-support exoskeletons (Figure 3), and the other concerns all other muscles involved in the trunk extension (TZ, LD, GM, and BF), in Table I.
DISCUSSION
- "This supports Abdoli, Agnew [25], who reported that increasing the load handled could reduce the benefits provided by passive systems."
I believe the effect will be more commonly the opposite: when using an exoskeleton, the percentage back muscle activity reduction (compared to the no-exo conditions) will be greater for a heavier weight.
This effect was observed when evaluating the Apex (Lamers et al., 2017) and the Robo-Mate (Huysamen et al., 2018) exoskeletons, where the authors evaluated within the same experimental campaign the muscle activity when lifting two different weights.
- "Overall, women had higher levels of muscle activity than men. This result may be explained by the fact that all participants carried the same load (15kg) in this study, whereas women generally have a lower strength than men, on average [18, 20]."
I think that it is sure that this is the explainantion.
Indeed, I suggest indicating this as a limitation of the study because you are making a comparison between tasks that may be close to the maximum effort for women while sub-maximum for men.
- "Generalization to the whole population seems therefore premature, even if the factor of sex has been included in the model."
I suggest writing "working population" which is the focus for this type of exoskeleton.
Finally, I suggest using the acronym LBP for low back pain as it is used for the name of the special issue. Also, MMH for indicating manual material handling is often used in similar studies.
Author Response
The authors would first like to warmly thank the reviewers for the time spent in reading our manuscript and providing comments that improve the text. We hope that the answers provided below and the amendments made in the manuscript (in red) will answer your concerns.
Reviewer 2
Comments and Suggestions for Authors
Dear authors, congratulations for the good work done.
I have just some minor concerns about the article.
INTRODUCTION:
References to MSDs and LBP incidence and their consequences need to be included.
Thank you for your comment. We added two references in the introduction (Line 36).
- Russo, F., et al., The Effects of Workplace Interventions on Low Back Pain in Workers: A Systematic Review and Meta-Analysis. Int J Environ Res Public Health, 2021. 18(23).
- Fan, X. and S. Straube, Reporting on work-related low back pain: data sources, discrepancies and the art of discovering truths. Pain Manag, 2016. 6(6): p. 553-559.
When citing a study use LastNameFirstAuthor + et al., + (PubblicationYear).
Thank you. We integrated this change into the text (Line 52 for example).
MATERIALS AND METHODS
Exoskeletons:
For the two active exoskeletons, it is not clear what really are all the different setting parameters and their defualt values (reactivity, softness, hold threshold parameters for the first device, sensitivity, reaction and counter force for the second).
"The reactivity as well as the softness parameters were set at 50% (default settings). The support level, level of assistance at the end of the movement, and hold threshold parameters were set to 20% (default settings)."
"The sensitivity, reaction and counter force parameters were set to 5 (default settings on a scale of 0–10)."
Both these sentences don't make sense for a reader who doesn't know these specific devices.
I suggest explaining in detail these settings only if you can provide a clear explanation (e.g., what sensitivity is, what the value 5 is).
So maybe it is better if you explain only the level of assistance.
Thank you for this feedback. Indeed, the explanation of the different settings for active exoskeletons may appear a little bit obscure to readers unfamiliar with these systems. There are many parameters and these are not necessarily identical from one system to another. And unfortunately we don't have more technical information on these parameters from the manufacturers. These are still black boxes in which users have no indication of the real effects and impact of their settings. However, in the methodology, we have chosen to mention them and to indicate the values set in order to ensure that this study can be reproduced by others or compared to others which use different settings.
Figure 2: Make it bigger and colored to make the exoskeletons more visible.
We have enlarged the image to make it more visible.
RESULTS
The EMG results are presented in two parts: one is specific to the ES muscles, mainly investigated by previous studies on back-support exoskeletons (Figure 3), and the other concerns all other muscles involved in the trunk extension (TZ, LD, GM, and BF), in Table I.
We have effectively chosen to present our EMG results in two parts to make them easier to read. The first concerns the data of the assisted muscles most studied in the literature (erector spinae), with a graphical representation. The second concerns the data of the other back muscles, in tabular form.
DISCUSSION
"This supports Abdoli, Agnew [25], who reported that increasing the load handled could reduce the benefits provided by passive systems."
I believe the effect will be more commonly the opposite: when using an exoskeleton, the percentage back muscle activity reduction (compared to the no-exo conditions) will be greater for a heavier weight.
This effect was observed when evaluating the Apex (Lamers et al., 2017) and the Robo-Mate (Huysamen et al., 2018) exoskeletons, where the authors evaluated within the same experimental campaign the muscle activity when lifting two different weights.
Thank you for your comment. Literature seems to show that passive exoskeletons have a limited level of assistance compared to active ones. In fact, these systems operate on the principle of storing and restituting energy using deformable materials such as elastics and springs to provide assistance. Their energy restitution capacity is therefore limited to the mechanical properties of the deformable materials. There is an optimal level of assistance which will depend on the total weight to be assisted (body weight and load). Below this level (for “low” weights to assist), the exoskeleton will perform well, but when the load exceeds the capabilities of the assistance system, the benefits observed can be reduced or even eliminated. For example, if an exoskeleton generates an assistance of 5kg and you have to lift 5kg and 20kg, in one case we can say that the assistance is 100% and in the other only 25%. Gains in terms of reduced muscle activity will then be impacted. Our reference to Abdoli et al. results was constructed in this way.
With regard to the study of Lamers et al., 2017, the APEX exoskeleton is a passive exoskeleton which does not seem to be affected by the level of load handled in their protocol. This result, which differs from other studies, could be related to the level of assistance of this exoskeleton which is probably not exceeded by the characteristics of the task studied (lower loads or equal to the assistance capacities). But without really knowing the specific assistance properties of each exoskeleton, it is difficult to decide on these considerations. In the second study (Huysamen et al., 2018), an active exoskeleton was used, what is different and could explain their results.
"Overall, women had higher levels of muscle activity than men. This result may be explained by the fact that all participants carried the same load (15kg) in this study, whereas women generally have a lower strength than men, on average [18, 20]."
I think that it is sure that this is the explainantion.
Indeed, I suggest indicating this as a limitation of the study because you are making a comparison between tasks that may be close to the maximum effort for women while sub-maximum for men.
Thank you. In this study, we chose not to normalize the load handled to the individual in order to get as close as possible to real working conditions (a given item that can be handled by either men or women). We added a sentence to incorporate your suggestion and our explanation. Line 564 "The results might have been different if the load handled had been adapted to each participant maximal force. However, with a same load for all, we are more representative of what can happen in real work situations”.
"Generalization to the whole population seems therefore premature, even if the factor of sex has been included in the model."
I suggest writing "working population" which is the focus for this type of exoskeleton.
Thank you very much for this point. We modified the sentence (Line 600): “Generalization to all workers seems therefore premature, even if the factor of sex has been included in the model”
Finally, I suggest using the acronym LBP for low back pain as it is used for the name of the special issue. Also, MMH for indicating manual material handling is often used in similar studies.
Thanks for this feedback. We have integrated it throughout the document.
Reviewer 3 Report
The reviewer would like to thank the authors for their nice contribution and efforts in understanding exosqueleton consequences on biomechanics.
Even if the paper is very well written and show very interesting results, some wonders rose when reading:
* in the analysis, the claimed effect of each exoskeleton would help understanding the results. It is done throughout the text for active exoskeletons, but not for the passive one. Maybe a specific paragraph could help.
* analysis of the results are made with time-independent values. Even if ROM are not influenced by the exoskeletons, the strategy during the movement should be different. Then, it is a little bit excessive to conclude that "the muscular benefits observed during the use of this system is therefore related to its assistance and not to any modification of the kinematics" (line 469-471) and it would be better to say that the metrics used didn't show any any correlation...
* line 559-562: not so clear ! Are the differences effective (statistically proven) and small ?
* As a general remark, it would be interesting to clarify the type of shoe used, as it changes all the gait especially for women that have heeled shoes more frequently.
* last, the study is on short-time effects. What about more long term effects ? Modifying the natural movement may induce musculoskeletal problem for example.
Author Response
The authors would first like to warmly thank the reviewers for the time spent in reading our manuscript and providing comments that improve the text. We hope that the answers provided below and the amendments made in the manuscript (in red) will answer your concerns.
Reviewer 3
Comments and Suggestions for Authors
The reviewer would like to thank the authors for their nice contribution and efforts in understanding exoskeleton consequences on biomechanics.
Even if the paper is very well written and show very interesting results, some wonders rose when reading:
In the analysis, the claimed effect of each exoskeleton would help understanding the results. It is done throughout the text for active exoskeletons, but not for the passive one. Maybe a specific paragraph could help.
Thank you for your feedback. It was not our intention to underline the results obtained with the passive system. Indeed, the topic of this article was to evaluate new technologies and in particular active systems. So in our study we compare active systems to a passive one, as done in the results on line 296-301; 319-331 and compared to literature in the discussion (paragraph 4.1).
Analysis of the results are made with time-independent values. Even if ROM are not influenced by the exoskeletons, the strategy during the movement should be different. Then, it is a little bit excessive to conclude that "the muscular benefits observed during the use of this system is therefore related to its assistance and not to any modification of the kinematics" (line 469-471) and it would be better to say that the metrics used didn't show any any correlation...
Thank you. We have considered your feedback and modified the sentence to avoid confusion. Line 482: “Thus, based on previous studies and our metrics used (average angle), the use of BackX does not seem to induce major changes in overall trunk kinematics. The muscular benefits observed during the use of this system is probably related to its assistance and not to any modification of the kinematics as suggested in the literature [15]. More in-depth studies of whole-body kinematics would be interesting to confirm these results.”
line 559-562: not so clear ! Are the differences effective (statistically proven) and small ?
These differences between men and women are statistically different, but they are few in number and observed for certain muscles and/or for certain tasks. In view of our results, it is not possible to say that an exoskeleton is better than another in terms of muscle activity reduction for a given population (men or women). However, we observed overall specificities between the sex of the participants and the model of exoskeleton used. We have modified this section to make it clearer. Line 573-580: “Moreover, the exoskeleton model induced some significant differences between men and women. However, these differences were not consistent across the different muscles and tasks and never influenced the erector spinae muscle activation. Thus, our results do not identify an exoskeleton technology or design that is more appropriate, regarding female or male workers. Nevertheless, it would seem that P-EXO (BackX) and A-EXO2 (CrayX) allowed women to stand out from men in reducing the activity of certain muscles in some tasks. In parallel, for men, P-EXO (BackX) and A-EXO1 (ExoBack) tended to be different, by comparison with women, for certain (other) muscles and (other) tasks."
As a general remark, it would be interesting to clarify the type of shoe used, as it changes all the gait especially for women that have heeled shoes more frequently.
In this study, all participants used safety footwear, which is recommended in companies for MMH. We therefore added a sentence in the section 2.2 to specify the footwear used. Line 141: “All participants wore safety shoes during the study.”
Last, the study is on short-time effects. What about more long term effects? Modifying the natural movement may induce musculoskeletal problem for example.
Thank you for pointing to this area of research. Indeed, it would be interesting to investigate the long-term effects, as there is currently no data on the contribution of the use of exoskeletons in a real work situation on reduction in the occurrence of MSDs. However, long-term assessment requires considering other parameters as the acceptance of these systems, the familiarization with them and their real use in the workplace. However, in view of the data collected in our study, whether at the level of muscular activities or kinematics, we can raise the question of the long-term effects, whether positive or negative. As you mention, the modification of the kinematics could for example induce constraints on the long term. A disorganization of muscular synergies linked to the assistance of certain muscles could also be damaging for the joint. This is a very interesting topic, with many assumptions for future researches but that was not the aim of the present study, which only focused on the initial short-term effects of active exoskeletons.